# Electrophysiology Methods for Assessing of Neurodegenerative and Post-Traumatic Processes as Applied to Translational Research

**DOI:** 10.3390/life14060737

**Published:** 2024-06-07

**Authors:** Rezeda Ramilovna Shigapova, Yana Olegovna Mukhamedshina

**Affiliations:** 1Institute of Fundamental Medicine and Biology, Kazan (Volga Region) Federal University, Kazan 420008, Russia; shigapova.r.r.7@gmail.com; 2Department of Histology, Cytology and Embryology, Kazan State Medical University, Kazan 420012, Russia

**Keywords:** electromyography, human, animal, neurodegenerative diseases, traumatic brain and spinal cord injuries

## Abstract

Electrophysiological studies have long established themselves as reliable methods for assessing the functional state of the brain and spinal cord, the degree of neurodegeneration, and evaluating the effectiveness of therapy. In addition, they can be used to diagnose, predict functional outcomes, and test the effectiveness of therapeutic and rehabilitation programs not only in clinical settings, but also at the preclinical level. Considering the urgent need to develop potential stimulators of neuroregeneration, it seems relevant to obtain objective data when modeling neurological diseases in animals. Thus, in the context of the application of electrophysiological methods, not only the comparison of the basic characteristics of bioelectrical activity of the brain and spinal cord in humans and animals, but also their changes against the background of neurodegenerative and post-traumatic processes are of particular importance. In light of the above, this review will contribute to a better understanding of the results of electrophysiological assessment in neurodegenerative and post-traumatic processes as well as the possibility of translating these methods from model animals to humans.

## 1. Introduction

The advancement of medical science and its interaction with new technologies facilitates both the emergence of completely new methods of examination and provides an opportunity to utilize traditional diagnostic methods. Electrophysiologic studies have long established themselves as reliable methods for assessing the functional status of the brain and spinal cord (B and SC), the degree of neurodegeneration, and evaluating the effectiveness of current therapies. Electrophysiology is a valid method to detect, monitor, and quantify neural tissue function including signal conduction such as neural processing and reflex testing using evoked potentials (EPs) [1].

Electrophysiological assessment allows for addressing differential and local diagnoses, predicting functional outcomes, and evaluating the effectiveness of ongoing therapeutic and rehabilitation programs not only in clinical settings, but also at the preclinical level [1]. Given the urgent need to develop potential stimulators of neuroregeneration, obtaining objective data when modeling neurological diseases in animals seems relevant. Thus, in the context of applying electrophysiological methods, not only the comparison of the basic characteristics of bioelectrical activity of the brain and spinal cord in humans and animals, but also their changes against the background of neurodegenerative and post-traumatic processes are of particular importance.

The rehabilitation of motor pathologies, injury prevention, and diagnosis are areas of study for scientists and clinicians where there is a need to turn to technology to improve the outcomes and economic benefits. There is an increasing reliance on evidence to support the efficacy of therapeutic interventions, their prevention as well as the diagnosis of disease [2]. Over the past few decades, there have been impressive developments in many fields, providing powerful quantitative approaches to instrumental assessments in neuroscience (EEG, etc.), biomechanics (inertial sensors, EMG), and cardiology (ECG, etc.), among others [2,3].

To date, among the electrophysiological methods for assessing neurodegenerative and post-traumatic processes, the most popular is electroencephalography (EEG) to measure the electrical activity of the Band electromyography (EMG) to record changes in the SC and peripheral nervous system (PNS). Because the brain is a complex structure with complex nonlinear dynamics, complexity studies using brain imaging data such as EEG, magnetoencephalograms, and functional magnetic resonance imaging are becoming more common [3]. The EEG method determines the bioelectrical activity of the brain and is very informative and the most accurate as it displays a complete clinical picture: the level and distribution of inflammatory processes; the presence of pathological changes in blood vessels; early signs of epilepsy; tumor processes; the degree of impairment of brain functioning due to pathologies of the nervous system; and the consequences of stroke or surgical intervention. With regard to neuromuscular monitoring, EMG is relatively the easiest to use with high measurement accuracy [4]. Advanced EMG technologies provide much more physiological information than a simple bipolar method. For example, the use of surface electrode arrays allows for the detection of a so-called “image” of surface EMG that changes over time [2].

To date, among the electrophysiologic methods used for assessment [2], in addition to the above, surface EMG and EEG data are non-invasive and relatively inexpensive, which is also an advantage of these methods. Overall, over the past few decades, electrophysiology has shown great promise as a measure for assessing and stratifying neurological diseases, predicting functional outcomes, and informing clinicians about the planning and outcomes of therapeutic interventions. The evaluation of neurodegenerative changes in diseases such as Parkinson’s and Alzheimer’s disease in animal models includes a fair number of examples. Lu J and Sorooshyari S.K. (2022) applied machine learning to identify EEG in a rat model of Parkinson’s disease at sleep and at rest [5]. Petrovic J et al. (2020) investigated prodromal local sleep changes in rat models of Parkinson’s disease as an assessment of the stage of disease progression [6]. Alzheimer’s disease in a rat model was presented in Hector A. et al. (2023), where the main focus was to study the changes in EEG markers during the sleep–wake transition as well as changes in general activity during each of the two states [7]. In addition, Dringenberg H.C. (2000) found that cholinergic and monoaminergic interactions contribute to EEG slowing and may act as an assessment of the degree of disease progression [8].

The use of EMG in a rat model was applied by Johnson W.L. et al. (2012) to assess the level of locomotion recovery after lateral hemisection of the mid-thoracic spinal cord [9]. The use of this technique is suitable for injuries in any part of the spinal cord. For example, a group of scientists led by Rana S et al. (2021) proposed the recording of diaphragmatic EMG in unanesthetized free-running rats with cervical spinal cord injury. The assessment of brain injury can take place using both EEG and EMG [10]. For example, Büchele F. et al. (2016) conducted a parallel EEG/EMG study in a rat model after closed head injury to assess the sleep–wake ratio and post-traumatic stability [11].

With the above in mind, in this review, we sought to address the existing spectrum of electrophysiological research methods used in the basic or preclinical research phase. We selected the main electrophysiological assessment approaches that are also available in clinical settings including EMG and EEG. The current review may contribute to a better understanding of the results of electrophysiological assessment in animal models of neurodegenerative and/or post-traumatic processes, especially the possibility of comparing the findings with clinical practice.

## 2. Electrophysiological Methods as Applied to Translational Research

### 2.1. Electroencephalography

EEG is a two-dimensional representation of neuronal activity in the brain, which has become quite accessible for use nowadays, and represents the average sum of electrical currents generated by a group of neurons located in the region of the recording electrode [12]. The obtained spectrograms are, to some extent, suitable for diagnosing neurodegenerative diseases and monitoring the effectiveness of treatment in various B lesions. For example, one effective method to monitor the therapy of certain cerebral diseases (such as traumatic brain injury (TBI), epilepsy, and epileptic syndromes) can be a dynamic study of cerebral activity, providing an assessment of the shift of EEG abnormalities in a positive or negative direction [13].

Since Hans Berger first systematically analyzed the electrical activity of the brain, the EEG method has gained prominence in neurology, making a significant contribution to neurological diagnosis and has also proven useful in the fields of neuropsychiatry and psychology. Due to the use of a single electrode placement system (10–20), it is possible to perform the study with minimal variation between patients. It is also possible to identify typical clinical EEG waveforms, trace the existing correlation with specific brain states (e.g., wakefulness, sleep) and specific pathologies associated with abnormal EEG waveforms. In addition, thanks to this method, using computerized analysis, it is possible to monitor sleep and certain pathological conditions, allowing for a more complete history for the patient. Another potential application of EEG is, for example, in brain–computer interfaces, where EEG signals are used to control prosthetic limbs [14].

Traditional EEG, recorded with open eyes at rest, is supplemented by the registration of vegetative, motor, and many other indicators in various functional states of a person. These indicators are based on the activity of synchronizing and activating systems of the B, which differ depending on localization [15]. This method of registration and the analysis of biopotentials is one of the most effective tools for the non-invasive analysis of B activity. Therefore, it is difficult to assess the general functional state without using EEG [13].

It is worth noting that the first EEG measurements were performed on animals, and the mentioned method of electrophysiological analysis is often used at the stage of preclinical studies [16]. In humans, routine EEG recording involves placing recording electrodes on the surface of the head. However, for animal studies, there is a practice of placing electrodes in superficial and/or deep structures of the brain. This allows for more accurate measurements because the recording electrodes are in direct contact with nerve tissue. Additionally, EEG recording in animals can be carried out continuously over a long period of time (a day or more), which allows for more complete information about brain activity to be obtained.

One of the tasks of EEG studies is related to the detection of certain forms of rhythmic activity. Traditionally, the EEG of a healthy person has several main frequency ranges: alpha (α)-rhythm, beta (β)-rhythm, theta (θ)-rhythm, and delta (Δ)-rhythm. These rhythms are divided into fast-wave (α- and β-rhythms) and slow-wave (θ- and Δ-rhythms) rhythms. There are clear correlations between the appearance of a particular frequency component on the EEG and the functional state of the organism [17].

As for model animals, especially mammals, they also exhibit total electrical activity in the form of the rhythms above-mentioned [18]. However, localization of the source of EEG effects in most animal species is more complicated compared to humans due to the smaller brain size and the presence of a more extensive subarachnoid space, which makes it difficult to use this methodology. It becomes more challenging to accurately localize the sources of EEG rhythms and interpret the data obtained [16].

In addition, standard electrode placement in rodents is not mandatory. For larger species such as non-human primates, the adopted position is desirable [19], as their brain anatomy and electrical activity are more similar to humans in size and structure, while in rodents, these parameters differ. Consequently, other methods such as electroencephalography with multi-electrode implantation are used to measure and record brain electrical activity in small laboratory animals.

One of the most represented rhythms is the sensorimotor α-rhythm, which has been found in a range of animal species, from small rodents to primates. The α-rhythm is thought to be related to the excitability of neurons in the somatosensory cortex [20]. Its occurrence in humans can be recorded during wakefulness at rest in occipital cortical areas with eyes closed and under conditions of complete physical relaxation and relative mental inactivity. Depression of the α-rhythm is possible in the forward direction when combined with the β-rhythm as a result of visual, auditory, tactile, and other somatosensory stimuli or mental actions [21,22].

To date, there is no unambiguous answer concerning the localization of the α-rhythm in most animal species including rodents, which is connected to difficulties in localizing this rhythm in practice. However, it has been possible to establish its frequency, which is in the range from 7 to 14 Hz and corresponds to the range in humans. In most mammals including humans, the amplitude of the α-rhythm is variable and often below 50 μV.

While the sensorimotor α-rhythm has been studied in more detail, less information is known about the β-rhythm. β activity is commonly categorized into two ranges: β-1, from 15 to 20 Hz, and β-2, from 21 to 40 Hz. Due to the wide continuous expression of peaks in the β-band, the categorization of these ranges is conditional, which applies to both humans and many mammalian species. In humans, the area of β-rhythm realization is the anterior central gyrus, but spread is also observed in frontal and posterior leads [22].

Regarding the localization of the β-rhythm in animals, no unambiguous answer has been found. However, studies show that the sensorimotor β-rhythm in animals is produced by the precentral motor cortex [23,24]. Other studies on spontaneous activity have concluded that the postcentral somatosensory cortex also generates the β-rhythm [25,26].

In animals, a widespread sensorimotor β-rhythm has been shown to occur independently of motor function when somatosensory attention is engaged [27]. In rat studies, it was found that the onset of movement often coincided with an increase in the β-rhythm, while maintaining a fixed body position was often accompanied by a decrease in the value of this range [28].

Not much is known about β-bands in rodents. Scientists have managed to generate this rhythm in vitro during an intracellular recording of rat somatosensory cortex slices using microelectrodes filled with potassium acetate [29,30]. The findings suggest that the β-rhythm functionally separates activity in the superficial layers of the somatosensory cortex, represented mainly by the θ-rhythm, from output pathways in the deep layers, consisting of the β2-band. The β-rhythm was not detected in the somatosensory cortex of rodents in vivo. This may be due to the fact that, compared with the sensorimotor α-rhythm, the analogous β-rhythm has a much smaller amplitude and possibly a lower degree of phase conservation. As a result, the β-peak in the power spectrum would be much less pronounced. Second, β activity may overlap with the sensorimotor α-rhythm [31]. Third, in rodents, the β-rhythm may be absent in somatosensory areas and occur only in the motor cortex [27,32,33]. In addition to rodents, the β-rhythm is found in primates, especially in higher frequency bands, and there is strong evidence that it plays a role in motor function [34,35].

In addition to the above rhythms, low-frequency rhythms are also found. These are represented by θ- and Δ-rhythms. The frequency of the θ-rhythm is located in the range from 4 to 12 Hz for all mammals [22]. The expression of the θ-rhythm in rodents is observed less frequently than α- and β-rhythms. It is mainly represented as a hippocampal θ-rhythm. Often, the θ-rhythm dominates the EEG of rodents such as rabbits, rats, and mice. It is thought to be associated with excitation and is generated by cholinergic mechanisms. In addition to the aforementioned animals, it is also found in dogs and cats, but to a distinctly lesser extent, and is virtually absent in monkeys and humans [36]. Fluctuations in the frequency of this rhythm are indicative of the spatial variation in the rat’s position in the environment [37] and the relationship between the frequency of the θ-rhythm and the rate of spontaneous or electrically evoked movements in rats [38].

In clinical settings, the study of Δ-rhythm estimation (1 to 4 Hz) in humans is rarely performed. In animals, however, the description of this range is more common. This is primarily due to the fact that EEG acquisition is often performed invasively by applying electrodes to the cerebral cortex in a medicated sleep state. Studies show that the Δ-rhythm in rats increases during deep sleep and decreases during wakefulness, which is comparable to data in humans [39].

In addition to the basic rhythms in EEG recordings, there are also quite underutilized ranges such as the gamma-rhythm. This rhythm is a signal in the frequency range above 40 Hz and often occurs during intense concentration, meditation, or study. As a consequence, this rhythm is quite specific in the study of neurodegenerative diseases, especially in animal models. According to the literature, the focus of the maximum activity of γ-rhythm in healthy humans is localized mainly in the posterior parieto-occipital regions [40]. In rats, this rhythm is most often registered in the hippocampus or dentate gyrus, and may be the result of exploratory activity [41]. In AD, brain network activity in patients exhibits a marked decrease in γ-band oscillations, a feature that has also been observed in animal models such as rats and mice [42]. A similar manifestation in the form of reduced γ-rhythm power has been observed in PD patients [43] as well as in animal models (rats) [44].

Thus, under normal conditions, EEG data in animals and humans are practically comparable (Table 1, Figure 1).

However, some differences can be traced that are related to both anatomical features such as the shape and structure of the brain and behavioral patterns. In particular, the fast-wave bands represented by α- and β-rhythms differ in localization in the brain in animals and humans. Human α-rhythm generation is mainly represented in occipital cortical areas. However, in most animal species, especially small mammals, the identification of the location is difficult due to the peculiarities of the structure. The low-frequency bands, consisting of θ- and Δ-rhythms, also exhibit some differences. In rodents, the θ-rhythm, unlike in humans, is represented and dominates in the form of hippocampal θ-rhythm. The Δ-rhythm is more frequently used in animal clinical studies due to the easier availability of these results, which are considered to be quite informative.

### 2.2. Electromyography

EMG is a diagnostic method based on the indices of the bioelectric activity of muscles, through which it is possible to assess the functional state of skeletal muscle tissue and peripheral nerves, namely, to determine the focus, degree of prevalence, severity, and nature of their lesions [55].

There are several methods of EMG study:(1)Stimulation (non-invasive method): This is carried out by placing electrodes on the skin at the location of the muscles under study and recording the bioelectrical activity in a large area of the muscle.(2)Needle (invasive method): This is carried out with the help of a needle electrode, which is inserted into the muscle tissue. It is used to study the performance of individual muscle elements with greater accuracy than the non-invasive method.

There are several types of EMG curves:(1)The curve registers rapid fluctuations of the potential. The frequency is about 50–100 Hz. At rest, there is very little electrical activity, and the amplitude is low. This kind of EMG is considered normal. Indicators in a person can vary depending on age, weight, and physical development.(2)Fluctuations in electromyography occur with a frequency of less than 50 Hz. This indicates a decrease in muscle conduction. Such changes are characteristic of neuropathy.(3)The EMG curve registers a significant decrease in the frequency of oscillations. On average, it reaches 4–10 Hz. These results indicate the pathology of the extrapyramidal system.(4)Conducting electromyography does not cause any oscillations. Signs of the bioelectric activity of muscles are not detected, neither in a relaxed state nor with artificial stimulation. A similar condition occurs in muscle paralysis. Another name for type 4 EMG is “bioelectrical silence” [56].

EMG is quite informative in terms of clinical practice. The field of electromyography covers both surface and intramuscular EMG. In this case, surface EMG gives a global view of skeletal muscle function and allows for the activity of individual motor units to be assessed. In addition, multichannel surface EMG makes it possible to study the peculiarities of the work of several muscle systems. The recording of individual fiber potentials, occurring invasively, summarizes the action potential of a motor unit. A wide range of neuromuscular disorders alter the waveform of a motor unit action potential in different but characteristic combinations, the interpretation of which is the domain of neurological diagnosis [57,58]. Surface EMG is often used for monitoring neuromuscular pathologies, monitoring neuromuscular changes, the prevention of occupational disorders, and occupational therapy. Information on muscle activation during movement or effort complements clinical assessment and provides insight into both functional changes and abnormalities. Measurement of muscle activation provides information on the fatigability of motor units, recruitment/deactivation, joint contractions, synergy, and indicates the effectiveness of the rehabilitation plan. Today, surface EMG provides a greater amount of physiologic information. For example, the use of surface electrode arrays allows for the detection of surface EMG “images” that change over time. These data provide indirect information about muscle strength, motor unit recruitment and deactivation strategies, the rate of discharge, muscle length, location of innervation zones, myoelectric manifestations of muscle fatigue, and other phenomena. Current applications of EMG, mainly surface EMG, concern physiological studies, treatment planning, evaluation of interventions, monitoring of neurological disorders, and control of prostheses and robots [2]. For example, surface EMG can be used to trigger or control a functional electrical stimulation device for partially paralyzed limb muscles [59]. In patients with acquired CNS lesions or degenerative diseases, surface EMG can be used to better understand the underlying mechanisms of gait impairment to aid in clinical decision making or rehabilitation pathways and as a marker of disease progression or intervention efficacy [60]. The same considerations apply to many other subsections of the field of neurorehabilitation, exercise physiology, occupational medicine, and sports medicine.

Thus, electrophysiologic data are of great importance for understanding the mechanisms of neurologic diseases and may have a significant impact on the development of new methods of diagnosis, treatment, and rehabilitation for these diseases.

When conducting EMG, both the arbitrary bioelectric activity of muscles at rest and during their activation and the bioelectric activity of muscles caused by stimulation (ENMG) are evaluated [61]. It is possible to assess the total summarized electrical activity of muscles by means of EMG. Conducting electrical stimulation of the peripheral nerve with the determination of the latency period, shape, amplitude, duration of evoked potentials (EP), and excitation propagation velocity (EPV) along motor and sensitive fibers is possible with the help of ENMG. This technique allows for the evaluation of the state of the nerve along its entire length, judgment about the type and degree of nerve damage, and determination of the state of axon terminals [62,63].

The M-response and H-reflex are considered to be the most informative indicators in EMG registration. The M-response is a direct muscle response consisting of the total action potential arising in a muscle due to electrical stimulation of its motor nerve. A decrease in the amplitude of the M-response may indicate an electrophysiological reflection of a decrease in the number of contracting muscle fibers in the muscle due to the demyelination of nerve fibers, various myopathies, and diseases associated with a decrease in the number of motoneurons.

The H-reflex represents the total action potential of the motor units of a muscle, appearing under slight, weak irritation of afferent nerve fibers by electric current. Absence or a decrease in the amplitude of the H-response may indicate pathological changes in the anterior horn structures of the spinal cord, afferent or efferent nerve fibers, and posterior or anterior spinal nerve roots.

The measurement of the M- and H-reflex responses elicited by electrical stimulation of the sciatic nerve is a common method to assess motor function recovery after peripheral nerve injury both in clinical settings and in animal models of neurotrauma [64]. In humans, there is a decrease in the amplitude of both the M-response and H-reflex action potentials during injury, indicating changes in motor units [65].

In rats, the amplitudes of both responses during sciatic nerve stimulation in the literature present fairly unambiguous results. Often, it is observed that the amplitude of these responses exhibits lower ranges with an increase in the latency period of both the M-response and H-reflex in traumatized animals [64]. However, when therapeutic approaches aimed at neuroprotection are carried out, the amplitude of the M-response can partially recover [66].

Thus, it can be noted that the EMG technique and the types of obtained curves will not differ significantly in humans and animals (Figure 2). However, it should be noted that in clinical conditions, the non-invasive method of conducting the study will be predominantly used, in contrast to animals, where the needle method of EMG stimulation is required. Despite this, no significant differences were found in the manifestation of the M- and H-response parameters. It can also be observed that the decrease in the amplitudes of these electrophysiological indices during peripheral nerve injury is common to both humans and model animals. Nevertheless, a more rapid recovery of indices to the level of norm is observed more often in animals.

### 2.3. The Limitations and Challenges Associated with Using Electrophysiology in Translational Research

In addition to the problems described in the text of the article, there are also some problems that may affect the translation of EEG results from animals to humans. Often, EEG recording in humans is conducted in a non-invasive manner. Electrodes can be placed on the skull and electrical activity is recorded at the locations of interest. There is a uniform protocol for electrode positioning on humans to allow for a comparison of the results obtained regardless of the investigator [75]. When recording on an animal model, especially rodents, electrode placement is based on stereotaxic coordinates and may have differences depending on the study. Additionally, the position of electrodes on a human allows for the possibility to evaluate cortical processes; in contrast, in an animal, there is an opportunity to evaluate changes in brain activity in deeper structures. Therefore, it can be said that in animals, local field potentials can be measured in the brain structure required by the researcher by placing electrodes in certain areas of the brain. More often than not, this results in higher voltage results in animals compared to humans (about 10 µV to 1 µV), which makes animal measurements more sensitive.

Additionally important is the fact that humans have the ability to perform the experiment with their eyes closed, which eliminates artifacts. However, animals do not have such a possibility due to the peculiarities of their behavior [76]. Conducting a recording under anesthesia also affects its quality.

It is also important to note the difference in the shape of the cerebral cortex in humans and animals, especially in the rat and mouse. These animals have a fairly smooth surface of the cortex while in humans, the cortical surface is characterized by crinkles and grooves. This feature can affect the quality of recording when it is necessary to search for the signal source from the skull surface. Thus, the presence of a strong signal in a certain area does not guarantee its presence near the location of the electrode itself [77].

Another fact related to the anatomical structure of the brain is also important. The position of the hippocampus in humans and model animals is different. In humans, this structure is located in the medial temporal lobe, which makes it difficult to measure the EEG of this structure (epileptic patients may be an exception). In rodents, however, the hippocampus is located below the cortical mantle and quite strongly influences the recording of signals in the cortex, especially when recording the signal from the animal’s head [78].

These problems can be a barrier to translating the results to humans. Several suggestions can be made to solve these problems. Initially, EEG should be used in conjunction with any method or methods, not just as an independent criterion for disease assessment. Regarding the type of electrode placement, a non-invasive electrode position is appropriate for humans. For animals, invasive electrode placement is preferred. It is imperative to clarify the differences between the frequency ranges of different species and to know the clear biomarkers of human EEG in different diseases.

Thus, additional studies including those with large samples such as genetic studies are needed to find more precise relationships between EEG and individual genes [79].

Problems in translating data from animals to humans are also encountered in EMG. In contrast to EEG, EMG acquisition is somewhat more difficult due to the large number of possible subjects. Depending on the expression of myosin heavy chain isoform in muscles, muscles are most often divided into slowly contracting (type I) and rapidly contracting fibers (type II) [80]. In humans, muscle groups are mostly a homogeneous mixture of fast contracting and slow fibers, whereas rodent muscle groups show a more pronounced division into type I or type II muscles with a predominance of fibers [81,82,83,84,85,86,87]. Otherwise, the characteristics of human and rodent locomotor models are very similar.

However, dynamic tasks are often the preferred condition for EMG measurements. In such cases, the experimenter faces a number of problems, as it is quite difficult to make the animals perform an isometric task while performing EMG recordings. Thus, the limitation of dynamic contractions will manifest as large motion artifacts contaminating the EMG signal, occurring when the electrode moves relative to the muscle during contraction [88,89], changes in the conduction properties of the muscle tissues [88], and from the electrode cable in wiring systems [90]. Although this is unavoidable when recording from both humans and animals, it can be minimized by using signal processing techniques. Furthermore, to better assess the significance of animal muscle data and extrapolate them to humans, it is necessary to have data on the biomechanics of organisms, in order to have the same recording methodology.

Thus, it should be remembered that each of the presented methods is not a stand-alone method for the assessment of certain diseases and their application should be comprehensive. In addition, research toward translational medicine should continue.

### 2.4. Methodological Features

Electrophysiologic studies are important tools for studying electrical activity. According to the literature review, the protocol items for both EEG and EMG may vary depending on the purpose. It is worth remembering that both EEG and EMG have an invasive and non-invasive method of imaging. However, there are general rules specific to each method. The first important thing is to prepare the subjects or model animals for the study. Patients should avoid caffeine and other excitatory substances, and follow sleep and nutritional guidelines. Proper electrode placement is also essential. Researchers should follow standard electrode placements according to international systems (e.g., 10–20 system for EEG) and perform impedance checks to ensure good electrode contact. In addition, it is worth remembering to control for interference. Electrophysiological studies are sensitive to various interferences such as electrical signals from external sources, muscle movements, and other artifacts. Researchers should take steps to control and minimize interference such as using shielded rooms and filters. To ensure the reproducibility of results, researchers should use standardized stimulus and task protocols. This allows for a comparison of results between different subjects and research groups. It is important to use proper methods and statistical approaches when analyzing the data obtained. Researchers should be alert to possible distortions in the data and apply correction methods where necessary. It is important to fully document all stages of the study including the methodology, protocols, data obtained, and equipment used. This will help to ensure transparency and the ability to replicate the results of the study.

Following these methodologies and protocols will help researchers ensure the reliability and reproducibility of results in electrophysiological studies. However, despite the above, there are some methodological papers as well as articles that can serve as a basis for conducting a study with both EMG and EEG [91,92,93,94,95,96,97,98].

### 2.5. Method of Local Potential Fixation and Microelectrode Arrays

To date, the EEG and EMG methods are sufficiently reliable and are not expensive to use directly on patients, neglecting the use of preclinical studies on model animals. However, it is worth considering that there is still no complete representation of the biomarkers of neurodegenerative and post-traumatic diseases, which in turn makes further research in this direction relevant.

Despite this, for a more complete preclinical study, it is possible to use a multidisciplinary approach that includes other methodologies in addition to standard electrophysiology methods.

One of the new directions is the study of model cells based on the method of local potential fixation (patch-clamp) or microelectrode arrays (MEAs) to better determine the cause of altered neuronal activity in neurodegenerative and post-traumatic processes.

Due to the patch-clamp method, our understanding of ion channels on cell membranes has grown significantly. Ion channels have been found to be important components for regulating osmotic pressure inside and outside cells and maintaining cell membrane potential [99]. ATP-sensitive potassium channels (K-ATP channels) are quite important channels linking membrane excitability to cellular metabolism and are widely expressed in smooth muscle cells, skeletal muscle cells, and nerve cells [100,101]. K-ATP channels belong to the family of internally rectifying potassium channels and are divided into sarcolemmal K-ATP channels (sarcK-ATP) and mitochondrial K-ATP channels (mitoK-ATP) [102,103]. These types of channels consist of inwardly rectifying potassium subunits (Kir 6.1 and Kir 6.2) and sulfonylurea receptor subunits (SUR 1 and SUR 2). They have been shown to be widely distributed in many brain regions including the hippocampus, hypothalamus, and basal ganglia [104,105]. However, the role of the different distribution and composition of K-ATP channels in neurodegenerative diseases is unknown. An important feature of these channels has been revealed: their activity is influenced by intracellular ATP levels and the ratio of ATP and ADP concentrations. Under normal, physiologic conditions, most K-ATP channels are closed and do not affect neuronal excitability. However, under certain stress conditions, the channels can be activated to regulate excitability, which affects many cellular functions [106,107]. Neurodegenerative diseases are most often accompanied by metabolic disorders, ischemia, cell necrosis, and hypoxia [108]. During ischemia and hypoxia in the brain, there is excessive release of glutamate, which further triggers excessive Ca^2+^ influx into postsynaptic neurons and leads to an overload of Ca^2+^ channels and aggravates neuronal death [109]. However, when ATP levels are reduced, the activation of K-ATP channels is also observed, which can cause membrane hyperpolarization. This may help to inhibit Ca^2+^ influx and reduce intracellular Ca^2+^ concentration [110]. In addition, mitoK-ATP-channels may reduce neuronal apoptosis and protect synaptic function by regulating oxygen free radical production during hypoxia and ischemia [111]. Given the role of K-ATP-channels in the regulation of neuronal excitability, synaptic transmission, neurotransmitter release, and plasticity, these channels are emerging as a new target in the treatment of neurodegenerative diseases [107,112]. K-ATP channel-opening agents have been developed to target K-ATP channels including benzopyrans, cyanoguanides, thiocarbamides, etc. [113].

Neurodegenerative diseases such as Alzheimer’s disease (AD) and Parkinson’s disease (PD) are characterized by an increased excitation of CNS neurons, associated with either a significant loss of GABAergic interneurons or dysfunction of interneurons due to the loss of their afferent excitatory input or changes in their receptors. An imbalance of neuronal inhibition and excitation is also found in many other neurological and psychiatric disorders [114]. In AD, symptomatic manifestations are marked by impairments in memory and cognitive function, which are thought to be due in part to hippocampal hyperactivity as a result of the loss of function of inhibitory interneurons [115]. Given that GABAergic interneurons play an important role in regulating hippocampal activation levels through inhibition, the processes involved in learning and memory formation require a balance of excitatory and inhibitory neuronal network activity [116]. Consequently, the loss or dysfunction of GABAergic interneurons can lead to learning and memory dysfunction. In rodent models, a significant decrease in the number of GABAergic interneurons is one of the notable changes observed in the aging hippocampus [117,118,119]. Alterations in the function of GABAergic interneurons result in increased neuronal activity in the CA3 region of the aged hippocampus, which contributes to memory dysfunction because overactive CA3 pyramidal neurons are unable to encode new information in a pattern typically seen in young adults [118,120]. This view is supported by the observation that the overexpression of NPY (a neuromodulator released by a subclass of GABAergic neurons) in the CA3 region of the hippocampus or treatment with low doses of valproate (a drug that increases GABAergic neurotransmission) can improve hippocampus-dependent long-term memory in old rats [117]. In addition, decreased GABA levels are more pronounced in apoE4 carriers (comprising 60–75% of AD patients), showing increased brain activity at rest and in response to memory tasks [121,122]. Moreover, apoE4 expression in mice causes an age- and sex-dependent (female > female) decrease in the number of GABAergic interneurons, which correlates with the severity of learning and memory deficits [123,124]. Thus, excitatory–inhibitory imbalance in the hippocampus likely contributes to memory dysfunction in both aging and Alzheimer’s disease [115]. From this perspective, strategies that improve network balance may prevent or reduce cognitive decline in aging and Alzheimer’s disease. These methods may include transplantation or drug replacement therapy.

In PD, motor deficits can be noticed in the early stages, which then develop into a diverse set of symptoms including changes in mood, sleep, and general cognitive dysfunction [125]. The treatment of PD has relied heavily on pharmaceutical drugs such as dopamine (DA) agonists to compensate for decreased DA concentrations in the striatum as a result of the loss or dysfunction of DAergic neurons in the substantia nigra of the midbrain. Dopaminergic cell transplantation is considered a good alternative to treatment with DA agonists, and multiple studies in animal models have demonstrated the efficacy of these transplants [125]. However, the results of DAergic cell transplantation in PD patients have been mixed with frequent side effects [126], prompting more attention to alternative therapeutic options [127]. Although DAergic cell therapy is still being considered as a treatment option, new results in animal models suggest a role for GABAergic transplantation in the treatment of PD. The use of medial ganglionic eminence (MGE) progenitor cell transplants for the treatment of motor symptoms has also gained promise. MGE cells were transplanted into the striatum and then the fate of the graft-derived cells and their effect on motor deficits were analyzed. Although the graft cell survival rate was low (~1%), the graft-derived cells differentiated into GABAergic interneurons expressing GABA. Moreover, the majority of graft-derived cells exhibited properties of inhibitory interneurons. More importantly, transplantation improved motor behavior, as evidenced by a decrease in apomorphine-induced rotations and an increase in stride length. It turned out that these cells also survived for up to a year after transplantation. Even with the modest cell survival rate, these results suggest that the transplantation of GABAergic cells deserves further evaluation as a method of restoring the necessary inhibitory function of damaged circuits in the brain in PD.

In addition to the patch-clamp method, the multi-electrode array (MEA) method is increasingly being used by researchers as an instrumental platform for monitoring long-term nondestructive recordings of spontaneously activating neurons in vitro for use in modeling PD, AD, schizophrenia, and many other human CNS diseases. This method allows for the monitoring of spontaneous electrophysiologic activity in in vitro neuronal cultures. With the wider use of these tools, there is a need to explore the state-of-the-art of research using MEA and to identify best practices for data acquisition and analysis to avoid errors in the interpretation of the data obtained. Using a dataset of primary cortical culture recordings from animals such as rats, statistical power was explored to detect changes in neuronal activity at the array level. In addition, the implementation of an unsupervised spike sorting method allows for the use of action potential incidents down to the single neuron level. After the implementation of spike sorting, the dynamics of firing frequency in populations of individual neurons and networks were investigated longitudinally. Finally, the ability to detect a frequency-independent phenotype, the change in action potential amplitude, was demonstrated by treatment with pore-forming neurotoxins.

MEAs consist of a large number (tens to hundreds) of planar electrodes embedded in the base of a tissue culture chamber that allow for the parallel detection of local field potentials generated by spontaneous or evoked neuronal excitation. MEAs measure potential differences at the recording and reference electrodes at high frequency (10–60 kHz), and action potentials (peaks) are detected when sample values deviate significantly from the background potential. Activity within cultures is quantified by the frequency of peak events, and in some applications, through the occurrence of synchronized peak events or “spikes” [128,129].

This method has quite a few positive attributes. MEAs are capable of recording over hundreds of channels simultaneously in a much less labor-intensive manner than traditional single-channel electrophysiological methods such as patch-clamp recording, albeit with a lower degree of spatial resolution. Because MEA recordings do not destroy cultures, repeated recordings can be performed as long as the integrity of the culture is maintained. For these reasons, the use of MEA is becoming increasingly common in neurobiology and biomedical research. However, methodologies in this field are still being developed. To date, the most comprehensive methodological review of in vitro MEA methods has been performed by Novellino et al. (2011) [130].

This method can also be used as a preclinical study to detect neurotransmitter release in real-time.

Thus, both methods can complement preclinical studies of diseases. The patch-clamp method provides high accuracy and signal-to-noise ratio for detecting single neurons, whereas MEA outperforms it by simultaneously detecting the electrophysiological activity of neuronal networks at multiple locations in a non-invasive and high-throughput manner.

## 3. Peculiarities of Electrophysiologic Markers in Neurodegenerative Diseases

### 3.1. Parkinson’s Disease (PD)

Decreased α-activity in both hemispheres at rest is one of the main EEG changes observed in patients with PD [12]. The presence of a correlation between the slowing down of the α-rhythm at rest and akinesia has been revealed. In some cases, the EEG of patients with PD shows a complete disappearance of the α-band and a decrease in the amplitude of β1- and β2-bands [131]. Under conditions of the chemical modeling of PD in animals, more often rodents, there is an increase in α-waves and a decrease in the frequency of the β-band on EEG [132]. Slow-wave oscillations of PD patients are accompanied by negative dynamics in the form of the increasing amplitude of Δ- and θ-bands. This is likely connected with disturbed mechanisms of brain regulation and indicates the impaired functioning of brain activating systems at the initial stages of the disease [133,134]. It is worth noting that the θ-rhythm dominates on the EEG of PD patients, especially in the later stages of the disease. In model mice with PD at rest, an increase in θ-frequency as well as Δ-rhythm in the cortex and striatum was noted [94,135]. Unfortunately, a deeper analysis of the presented rhythms in rats and mice against the background of PD modeling has not been performed. Interhemispheric asymmetry, desynchronization, and increased motor artifacts during tense wakefulness are also found in PD patients [136], which cannot be assessed in small laboratory animals.

The EMG method in humans with PD is often used as an additional diagnosis. There is a growing interest in developing wearable systems to objectively monitor the motor fluctuations of patients during their daily activities outside the clinic [137].

Most of these systems use motion sensors to provide information on early markers of PD manifestations including recording dyskinesia, tremor type, motor fluctuations, and gait characteristics [138,139].

In patients with PD, bradykinesia is accompanied by a delayed onset of muscle activation and an abnormal EMG pattern during movements; rigidity results in a large number of late EMG bursts after the cessation of voluntary muscle contraction and increased resistance to passive stretching [140].

In tremor, EMG bursts usually have an alternating pattern. This approach is practically not used in model animals given the use of the chemical induction of PD, leading to the rapid development of the symptomatic stage of this disease. Conducting EMG on model animals with PD manifestations is accompanied by the absence or minimal activity of EMG parameters [94,141], with their more abrupt fading with disease progression in contrast to humans, where changes in electrophysiological activity are less pronounced and have a pro-gradient course [141].

Thus, it can be seen that for model animals with PD, unlike humans, the increase in high-frequency bands on the EEG is characteristic. At the same time, a decrease in the indicators of slow-wave rhythms is preserved in both animals and humans.

The use of EMG in patients with PD is possible as a diagnostic tool to search for early markers of this disease. However, in model animals, particularly rats, in chemically induced modeling of the disease, the use of EMG will not be reasonable due to the rapid development of the symptomatic stage of the disease.

### 3.2. Alzheimer’s Disease (AD)

The main EEG changes in human AD are associated with a general slowing down and reduction in α- and β-rhythms, resulting in increased Δ- and θ-activity [142]. In the chemical modeling of AD in rats, a decrease in α- and β-wave frequency was also observed [143]. Additionally, an increase in the cortical slow θ-rhythm and functional slow coupling θ-activity in parieto-occipital regions, along with an increase in Δ-rhythm power, has been found in animals with AD [144]. Moreover, a decrease in α-activity in rats correlates with a decrease in the Δ-range. Additionally characteristic for rodents in AD is a slowing of neocortical EEG, indicating a reduction and desynchronization of EEG patterns [8]. Furthermore, a decrease in higher frequency components of EEG in the occipital and temporal regions of AD patients correlates with cognitive decline and disease severity [16,132,145].

Changes in the frequency and amplitude of EMG rhythms in AD are similar to those in PD, as extrapyramidal signs such as rigidity, bradykinesia, shuffling gait, and posture changes are relatively common in these neurodegenerative diseases [146]. However, patients with AD have higher cortical excitability (i.e., lower motor thresholds at rest and during activity), lower cortical inhibition, and impaired cortical plasticity [147]. These changes are also characteristic of mammals. In model animals, particularly rats suffering from AD, there may be changes in muscle activation, which may manifest as a decrease in amplitude or a change in signal frequency, and an increase in response latency time between stimulus and muscle response. 

Summarizing the above, we can conclude that the EEG and EMG parameters in model animals and in humans with AD are similar. There is a general increase in high-frequency and a decrease in low-frequency ranges of EEG, along with a decrease in the amplitude and frequency of the EMG signal.

## 4. Peculiarities of Electrophysiologic Markers in Traumatic Injuries of the Central Nervous System

### 4.1. Traumatic Brain Injury

Traumatic brain injury is an acquired brain injury caused by external mechanical force that can lead to both minor and major consequences in the functioning of neural tissue [148]. The EEG imaging procedure has been proposed as a clinical diagnostic test to detect, confirm, measure, and localize traumatic brain injury in patients. There are no clear-cut EEG features unique to TBI [149]. Additionally, EEG changes are not uniform among patients due to differences in injury severity, and stabilization of the encephalogram in some cases is possible as early as 15 min or several hours after concussion [150].

After a traumatic brain injury, epileptiform activity (sharp waves of high amplitude or high-frequency discharges) often occurs, followed by diffuse suppression of cortical activity, usually lasting 1–2 min, and then diffuse slowing of the EEG, which typically returns to normal within 10 min to 1 h [151]. Quantitative EEG in patients most commonly shows an immediate decrease in the mean α-frequency [152], an increase in the θ-rhythm waveform [153] Δ-activity [154], and in the ratio between the θ- and α-frequencies [155,156]. In rats, TBI leads to a decrease in the indices of the above rhythms, but in mice during wakefulness, an increase in the θ- and Δ-bands has been observed with this type of neurotrauma [157]. Fluctuations in the amplitude of the α-rhythm in healthy rats increase as they approach the occipital cortex, while in PMT, it decreases in all leads. Regarding the β-rhythm, the literature also describes a decrease in its frequency and amplitude in rats with traumatic brain injury.

Increased δ-rhythm power and decreased α-rhythm predominance in the normal awake state have been described in both acute [154] and subacute [158] trauma survivors. In a rat model of traumatic brain injury, similar changes occur, associated with an increase in the amplitude of δ-rhythms in all leads. A few weeks or months after traumatic injury, patients show an increase in the frequency of the posterior α-rhythm by 1–2 Hz, which is explained by a return to baseline after post-traumatic deceleration [149]. Rats show a more rapid recovery of EEG parameters after traumatic brain injury than humans. One of the possible reasons may be the high ability of rat tissues and organs to regenerate [159].

Thus, in patients with traumatic brain injury, we observe a widespread decrease in the frequency of fast-wave bands, particularly the α-rhythm, while the power of the other bands increases. On the contrary, in small laboratory animals, the δ-rhythm increases with a general decrease in the amplitude and frequency of other EEG rhythms during the modeling of traumatic brain injury. It is also worth noting that changes in the θ-/β-rhythm ratios are more characteristic of patients with traumatic brain injury, whereas for rats and mice, it is the θ-/α-bands.

### 4.2. Spinal Cord Injury (SCI)

While clinical assessments offer a comprehensive analysis of sensorimotor function, electrophysiological methods identify neuroanatomical and physiological indicators of the peripheral nervous system (PNS) and central nervous system (CNS) to assess functional prognosis [160]. For instance, relying solely on the American Spinal Injury Association (ASIA) impairment scale as the primary outcome for SCI clinical trials poses challenges, as numerous conversions to the ASIA degree may not accurately reflect changes in the severity of neurological deficits [161]. In complementing the assessment of clinical outcomes, electrophysiological recordings have the potential to enhance SCI diagnosis and patient stratification [162,163]. The literature suggests that EEG could serve as a biomarker of functional recovery in SCI patients due to changes in cortical plasticity, which may impact EEG fluctuations. For example, patients with SCI have exhibited a decrease in α-rhythm power and an increase in β-frequency [164]. However, the reliability of using EEG for diagnostic or prognostic purposes appears to be inadequate. Consequently, employing EEG in the study of SCI in model animals also seems unnecessary.

The most reliable method for diagnosing SCI may be EMG, which serves as a crucial tool for studying motor responses after injury. Currently, the measurement of EMG responses is frequently utilized in clinical practice. In patients with SCI, M- and H-responses are often overlooked due to their limited informativeness. However, for model animals with SCI, the registration of these parameters is essential. Observations have shown an increase in M-wave amplitude alongside a decrease in the H-reflex [165,166].

In patients with SCI, motor evoked potentials (MEPs) are commonly analyzed to assess the level of damage and evaluate the functional capacity of the nervous system. According to the literature data, after SCI, the MEPs’ amplitude and reflex activity in individuals are typically absent for 14 weeks. Subsequently, during rehabilitation, the MEPs’ amplitude can gradually increase over the next 4 weeks and stabilize by 22 weeks in cases of mild injury with preserved paraparesis [167]. In animals, there is a marked decrease in the amplitude activity of the MEPs depending on the severity of the injury, but also a quicker recovery of motor functions post-injury [168]. Additionally, an increase in the latency period frequency of the MEPs in rats compared to the amplitude values has been observed [165].

It has been demonstrated that in SCI, repeated and prolonged muscle stretching leads to a decrease in the amplitude of the MEPs during maximal voluntary contraction in humans [169]. In rats, stretching induces EMG patterns similar to those observed in humans [170]. This suggests an equivalent effect of stretching on the nervous system in both humans and animals.

Therefore, the routine use of EEG as a diagnostic method for SCI in both humans and animals is not feasible. However, EMG analysis is often necessary in clinical conditions, especially in the delayed period, to address issues related to the preservation of conductive pathways and electrostimulation during neurorehabilitation. It is noteworthy that the EMG indices after SCI in humans and model animals do not show significant differences. Both humans and rats, and to a lesser extent pigs, exhibit a decrease in the amplitude of M- and H-responses and MEPs after SCI.

## 5. Conclusions

Despite the active integration of EEG and EMG in research involving model animals, it is important to acknowledge their limitations. Several factors should be considered when employing EEG: the necessity of mitigating the influence of external electromagnetic fields, challenges in implanting electrodes, and subsequent data imaging. A notable characteristic of EEG is its high sensitivity to various movements, even slight ones as well as the psycho-emotional state of the test animal. These factors manifest as artifacts in the recordings, impeding data analysis. Moreover, EEG has relatively low spatial resolution, making it suitable for studying neuronal response speed but less informative for pinpointing activity locations. Another challenge stems from EEG equipment being optimized for human heads, leading to difficulties in applying this technique to animals with significantly different skull shapes.

In animal studies, needle electrodes are often used, connected to cable leads, which can introduce artifacts due to pain receptor irritation, particularly during movement. Similarly, EMG using needle electrodes poses challenges including the need to secure the signal source and the transmission of electrical impulses through muscles, carrying risks of muscle and nerve trauma as well as infection.

There are no risks or complications associated with non-invasive EEG for either the patient or the model animal. Regarding invasive EEG, the most common complications that can result from an EEG are hemorrhagic complications. Some researchers have also described the manifestation of persistent neurological deficits after the procedure [171]. Most of the complications occur as a result of improper implant placement or its malfunction [172].

However, most of the potential risks of EMG are related to hemostasis disorders, the transmission of bloodborne pathogens, and the patient’s electrical sensitivity [173].

In conclusion, despite these drawbacks and considering species-specific differences, electrophysiological studies remain valuable in translational research. They enable the analysis of brain and spinal cord function, the degree of neurodegeneration, and therapy effectiveness assessment. EEG and EMG can aid in understanding the therapeutic intervention mechanisms related to neuronal plasticity, axon growth, and remyelination. However, refining electrophysiological analysis methods is crucial to obtain more accurate results in both clinical and experimental contexts. Utilizing advanced technologies such as multichannel electrodes and optical tomography, alongside machine learning algorithms and automated data processing techniques, can enhance the accuracy and objectivity of the analysis. Standardizing methods will further improve the quality and facilitate comparative analyses across different research centers.

## Figures and Tables

**Figure 1 life-14-00737-f001:**
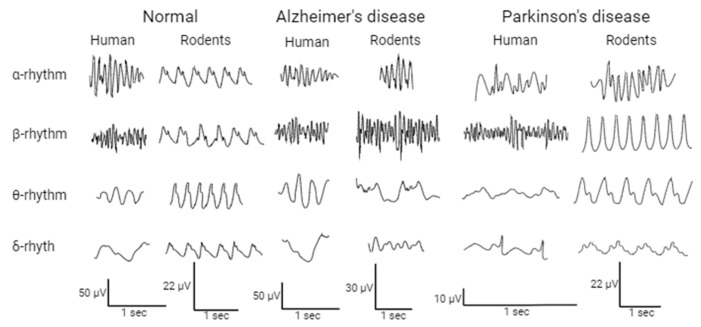
Comparison of EEG rhythms between humans and model animals. Alzheimer’s disease: Human [45], Rodent *α-rhythm, β-rhythm* [46], *θ-rhythm* [47], *δ-rhythm* [48]; Parkinson’s disease: Human *α-rhythm* [49], *β-rhythm, θ-rhythm* [50], *δ-rhythm* [51]; Rodent *α-rhythm* [52], *β-rhythm, θ-rhythm, δ-rhythm* [53]; Normal Human *α-rhythm, β-rhythm*, *θ-rhythm, δ-rhythm* [54]; Rodent *α-rhythm, β-rhythm, θ-rhythm, δ-rhythm* [53].

**Figure 2 life-14-00737-f002:**
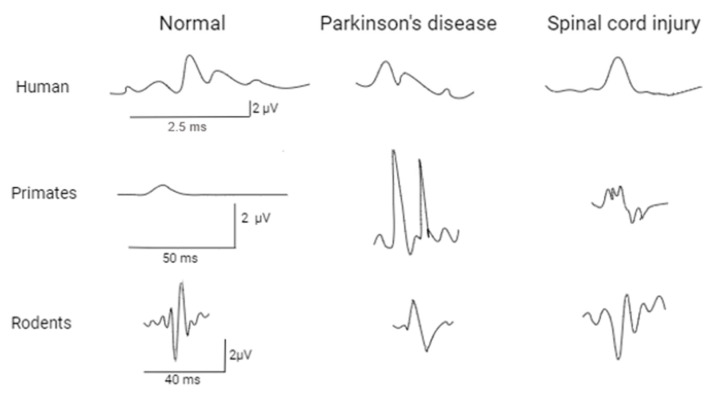
Comparison of somatosensory evoked potentials of EMG between humans and model animals. Spinal cord injury: Human [67], Rodents [68], Primates [69]; Parkinson’s disease: Human [70], Rodents [71], Primates [72]; Normal Human [73], Rodents [68], Primates [74].

**Table 1 life-14-00737-t001:** Comparison of EEG characteristics in different animal species and humans.

Species	Dominant Rhythm	Localization of Rhythms in the Brain	Features of Shooting Rhythms
Human	*α-rhythm.* Occurs with eyes closed in a state of complete rest and relaxation. *β-rhythm.* Associated with increased attention, concentration, and mental activity, often during complex problem-solving. *Θ-rhythm.* Most commonly observed in infants, and in adults during drowsiness, awakening, hypoxia, and shallow anesthesia. Δ*-rhythm.* Present during deep sleep without dreams.	*α-rhythm.* Occipital regions. *β-rhythm.* Anterior central gyrus. *Θ-rhythm.* Temporal areas, hippocampus. Δ*-rhythm.* Frontal areas.	- Conducted non-invasively; - The person should be well-rested, adequately fed, and in a calm state.;
Primates	*α-rhythm.* In a state of complete rest and relaxation, particularly with eyes closed. *β-rhythm.* Present during wakefulness and associated with physical and cognitive activity. *Θ-rhythm.* May be associated with the REM sleep phase and certain forms of brain activity. Δ*-rhythm.* Increased during deep sleep.	*α-rhythm.* Occipital cortical leads. *β-rhythm.* Anterior central gyrus. *Θ-rhythm.* Frontal-parietal leads Δ*-rhythm.* Frontal regions.	- The ability to use similar methodology for EEG acquisition in humans; - In some cases, EEG imaging in primates may require training the animal to remain calm and cooperative during the study.
Pigs	*α-rhythm.* Present during mental and physical rest and complete relaxation. *β-rhythm.* Present during wakefulness and muscular and cognitive activity. *Θ-rhythm.* Present during REM sleep and perception of new surroundings. Δ*-rhythm.* Present during deep sleep and recovery.	*α-rhythm.* In occipital and parietal cortical areas. *β-rhythm.* The cerebral cortex. *Θ-rhythm.* Hippocampus. Δ*-rhythm.* Frontal regions.	- Registration is often performed invasively in the medicated sleep state; - Determination of the areas of electrode placement is similar to the technique in humans. - Side effects regarding inflammation around implanted electrodes.
Rodents	*α-rhythm.* In the state of complete rest. Highest representation when eyes are closed. *β-rhythm.* In the waking state during activation of attention, concentration, problem solving. *Θ-rhythm.* When changing the position of the body in the environment, the phase of REM sleep, activity. Δ*-rhythm.* In deep sleep, anesthesia.	*α-rhythm.* Somatosensory cortex *β-rhythm.* Precentral motor cortex. *Θ-rhythm.* Hippocampal region. Δ*-rhythm.* Frontal areas.	- Recording is often conducted invasively in a medicated sleep state; - Identification of the areas of electrode placement is difficult; - Side effects regarding inflammation around implanted electrodes; - Social isolation is required due to the external components of the implanted EEG system.

## Data Availability

Not applicable.

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
