# Peer review of "Electrophysiology Methods for Assessing of Neurodegenerative and Post-Traumatic Processes as Applied to Translational Research"

_life, 2024, doi:10.3390/life14060737_

Round 1
Reviewer 1 Report
Comments and Suggestions for Authors
In this review article, the author explores and evaluates the electrophysiologic methods for assessing neurodegenerative and posttraumatic diseases. This manuscript will interest neuroscientists who perform electrophysiology. However, some issues need to be addressed.
Line 68: As a review, the author should discuss the high-frequency gamma (γ)-rhythm in EEG sessions.
Line 171: List the critical references of Table 1 and Figure 1.
Line 247: List the essential references of Figure 2.
Line 413: Is there any risk of an EEG or EMG?
Line 54: Correct the most popular “are” electroencephalography to “is.”
Line 55: The author should provide the full names of “GM” and “CM.”
Line 67: Correct the repeated number “2.”
Line 101: Change the verb from “is” to “are.”
Line 113: Replace " Its " with “It’s.”
Line 374: Replace “AIS” to “ASIA.”
Comments on the Quality of English LanguageNo comments.
Author Response
Dear Reviewer,
Please see the responses to your review in the attachment.

Reviewer 2 Report
Comments and Suggestions for Authors
The authors address the use of electrophysiological methods in animal models to study neurodegenerative and posttraumatic processes. The significance of this research lies in its potential to bridge the gap between basic research and clinical practice by providing insights into the functional state of the brain and spinal cord, neurodegeneration, and therapy effectiveness. The novelty of this review is in its comprehensive analysis of existing electrophysiological research methods and their application in translational research, highlighting the importance of comparing findings with clinical data.
The paper offers a thorough review of electrophysiological methods, emphasizing their relevance in studying neurodegenerative and posttraumatic processes. The authors provide a clear and concise overview of the current state of research in this field, highlighting the potential of electrophysiology in translational studies. The review also presents a well-structured argument for the importance of using electrophysiological techniques to assess brain and spinal cord function, making it a valuable contribution to the field of neuroscience and translational research.
I have only minor concerns as follows:
1. Provide more specific examples of studies where electrophysiological methods have been successfully applied to assess neurodegenerative and posttraumatic processes in animal models.
2. Include a section discussing the limitations and challenges associated with using electrophysiology in translational research, along with potential strategies to overcome these limitations.
3. Consider expanding the discussion on the implications of electrophysiological findings for clinical practice, including how these findings can inform the development of new therapies.
4. Incorporate more recent references to support the relevance and timeliness of the review, particularly in the context of advancements in electrophysiological techniques and their applications.
5. Clarify the criteria used to evaluate the effectiveness of therapy in the context of electrophysiological studies, including specific parameters or outcomes that are commonly assessed.
6. Provide a more detailed explanation of the methodologies used in electrophysiological studies, including any standard protocols or best practices that researchers should follow to ensure the reliability and reproducibility of their findings.
Author Response

(The authors gave the same response as above.)

Reviewer 3 Report
Comments and Suggestions for Authors
I am a little perplexed about the peer-review of this manuscript.
The claim of the manuscript is to summarize the electrophysiological methods used to determine neurodegenerative processes in animal models for the purposes of translational medicine applications. In light of this fact, I believe that the manuscript represents a good example of an old-fashioned essay (overall well-written) whose concept is a bit dated and nowadays of less utility.
The authors should modify the claim by making the manuscript a historical survey (which has already been well done by others as demonstrated by ref n.5 cited in the reference list). Otherwise, the Authors are invited to update the content of the review towards other electrophysiological techniques currently much more used for preclinical studies on animal and cellular models.
It is a fact that EEG and EMG techniques nowadays are very reliable and can be conducted directly on patients without any necessity of preclinical employment on animal models in order to assess further biomarkers of disease.
In addition, it must be underlined that the multidisciplinary "model-ephys-pharma-neurotherapeutics" approach is based mainly on other ephys methodologies, e.g. patch-clamp and MEA, rather than EEG and EMG. These are addressed to model cells or reduced preparations from transgenic animals in order to better address the origin of altered neuronal activity to further channelopathies or secondary channel dysfunctions (one for all, e.g. the reporting patch-clamp literature addressed to the study the pathophysiology of neuronal excitability in polyglutamine disorders.
Along this way, also EEG studies in neurodegenerative diseases focus on the electrophysiological activity of ionic channels (e.g. Shibuya K, Misawa S, Uzawa A, et al Split hand and motor axonal hyperexcitability in spinal and bulbar muscular atrophy Journal of Neurology, Neurosurgery & Psychiatry 2020;91:1189-1194).
Therefore, Authors limited their discussion to EEG and EMG techniques only, but even in this case the content is not updated. In fact they do not cite any reference dealing with new approaches within the EMG analysis applied to neurodegenerative disease like, e.g. the Nerve Conduction Studies (NCS), the Compound muscle action potentials (CMAPs) analysis, and MUNe and MUNIX methods (e.g., Kuwabara S, et al. Dissociated small hand muscle atrophy in amyotrophic lateral sclerosis: frequency, extent, and specificity. Muscle Nerve. 2008 Apr;37(4):426-30. doi: 10.1002/mus.20949.; Gaweł M. Electrodiagnostics: MUNE and MUNIX as methods of estimating the number of motor units - biomarkers in lower motor neurone disease. Neurol Neurochir Pol. 2019;53(4):251-257. doi: 10.5603/PJNNS.a2019.0026.).
As a whole, I warmly advise Authors to change their title reporting EEG and EMG acronyms (i.e., EEG and EMG electrophysiology methods for …..) as they limit their discussion to these two techniques only. Furthermore, I would consider whether to give mainly a historical slant or to update the content to a greater methodological contemporaneity. In any case, the content must be enriched and/or extended to other models and/or disorders.
Author Response

(The authors gave the same response as above.)

Round 2
Reviewer 3 Report
Comments and Suggestions for Authors
Dear Editor,
I evaluate an "Accept with minor revision" as the Authors did good improvements, as reported in the cover letter, although they forgot to insert them in the revised text. Once corrected accordingly the paper can be fully accepted. Also citations and references needs an accurate check-out.
Best,
Carlo Musio
Author Response
Уважаемый рецензент,
Пожалуйста, ознакомьтесь с ответами на Ваш отзыв во вложении.
